# Fast Terminal Sliding Mode Fault-Tolerant Control for Markov Jump Nonlinear Systems Based on an Adaptive Observer

**Pu Yang \*, Ziwei Shen, Yu Ding**  **and Kejia Feng**

Department of Automation, Nanjing University of Aeronautics and Astronautics, Nanjing 211106, China
\* Correspondence: ppyang@nuaa.edu.cn

**Abstract:** In this paper, a new adaptive observer is proposed to estimate the actuator fault and disturbance of a quadrotor UAV system with actuator failure and disturbance. Based on this, a nonsingular fast terminal sliding mode controller is designed. Firstly, according to the randomness of faults and disturbances, the UAV system under faults and disturbances is regarded as one of the Markov jump nonlinear systems (MJNSs). Secondly, an adaptive observer is designed to simultaneously observe the system state, fault, and disturbance. In order to improve the precision, the fast adaptive fault estimation (FAFE) algorithm is adopted in the adaptive observer. In addition, a quasi-one-sided Lipschitz condition is used to deal with the nonlinear term, which relaxes the condition and contains more nonlinear information. Finally, a nonsingular fast terminal sliding mode controller is designed for fault-tolerant control of the system. The simulation results show that the faults and disturbances can be observed successfully, and that the system is stochastic stable.

**Keywords:** fault-tolerant control (FTC); nonsingular fast terminal sliding mode control (NFTSMC); UAV; FAFE; Markov jump nonlinear systems (MJNSs)

## 1. Introduction

Markov jump systems (MJSs) were firstly proposed by N. M. Krasovskii and E. A. Lidskii in 1961 [1]. Over the years, relevant theories have been continuously improved. Since they can better describe the system of stochastic mode jump, MJSs have been gradually proven valid in the practical [2–4], and have had considerable research and application in the fields of economics, physics, unmanned systems, machine learning, and so on. On the other hand, unmanned aerial vehicles (UAVs) first appeared in the 1920s. Due to their outstanding performance on the battlefield, Western countries ushered in an upsurge of UAV research in the 1990s. In recent years, more and more cases of UAVs being used in rescue and disaster relief have sprung up [5–8]. Considering the safety and stability of UAVs, they can replace human beings by going to dangerous disaster relief sites and accomplishing some dangerous tasks. Due to the particular working environment, a quadrotor UAV system is easily affected by the wind environment, carried objects, human intervention, and other factors in the process of performing missions. Therefore, the output of the UAV system could be unstable. This kind of fault can be regarded as a Markov jump process, and the UAV system can be described by the continuous-time MJSs.

At present, there have been many studies on the stability and control law of Markov jump systems. Guan studies the stability of T-S fuzzy Markov jump systems based on sampling control in [9]. Wang studies the stochastic stability of the MJSs, which are affected by parameter uncertainty and actuator saturation [10]. Instead of asymptotic stability, Chen et al. [11] paid more attention to the changes in the transient properties. They studied the finite-time stability of a class of disturbed MJSs with random time delay. In terms of the reinforcement learning of agents, Jiang creatively combines the reinforcement learning method with Markov jump nonlinear systems (MJNSs). Based on this, he realizes optimal tracking control for MJNSs in [12], which opened new data-based fields in the studies

of MJSs. He et al. [13] utilized another reinforcement learning method with the optimal control for Markov linear jump systems.

Furthermore, fault-tolerant control (FTC) has been developed and is involved in industrial systems. Usually, FTC can be divided into passive and active FTC. Sliding mode control (SMC), which was proposed for a class of control problems with unknown disturbance, is one of the primary methods for FTC. Since it owns the advantages of quick response, and easy calculation and design, SMC has been used to design controllers and observers in fault diagnosis. For passive FTC, Liu et al. [14] study a novel SMC of a classic uncertain stochastic system with time delay, and design the corresponding adaptive sliding mode control law. Two experiments demonstrate the advantages of this method. Besides, for active FTC, Le et al. [15] proposed an extended state observer to estimate the fault of a robot manipulator. They designed a fault-tolerant conventional sliding mode controller and proved its stability. Considering a system with uncertain disturbances and actuator faults, Mao et al. [16], who combined the adaptive method with SMC, proposed a novel fault-tolerant controller to deal with the unknown bound of the input uncertainty. For nonlinear systems, Guo et al. [17] combined SMC with a radial basis function neural network and proposed a novel FTC control scheme. Zhao et al. [18] designed a novel nonsingular terminal sliding mode controller (NTSMC) for a quadrotor affected by variable mass. Compared with other SMC or FTC methods, the new FTC performs better.

Meanwhile, observer based on fault diagnosis and fault-tolerant control is a significant active FTC method. This method monitors the system's original state and actual state by establishing an observer system to observe the type and time of fault, and provide an essential reference for subsequent maintenance. A disturbance observer [19] combined with neural networks, FO calculus and SMC is utilized to approximate nonlinearities, actuator faults, and so on. For a Surface Vehicle, Wang [20] investigates a finite-time observer to design FTC to handle input saturations and uncertain faults. For multi-agent systems, an adaptive observer is designed for an event-triggered FTC to compensate for the fault in [21]. Song et al. [22] proposed an adaptive hybrid fuzzy output feedback controller based on a fuzzy observer to estimate the system state. A novel composite adaptive disturbance observer used to estimate the disturbances and faults is given to design the FTC in [8].

However, the fault-tolerant theorems are seldom used to control Markov jump systems. Scholars have concentrated on this field and done some research. Considering possible multiple faults in high-speed trains, a novel disturbance observer is given in [23] to promise the system's stability. Yang et al. [24] regarded a specific aero-engine system as one of the MJSs. Then, they considered that under the conditions of unknown sensor fault, actuator fault, and bounded external disturbance, a linear generalized reduced-order observer is designed to realize fault estimation, and the accuracy and effectiveness of the algorithm were validated. Adaptive and fuzzy theorems are utilized in the FTC of MJNSs in [25] to handle additive and multiplicative faults. For networked control systems, Bahreini et al. regarded them as classic MJSs. In consideration of the difficulty of estimating stochastic fault, a novel auxiliary system approach is used to estimate the random fault of continuous-time MJSs in [26]. In the case of partly unknown transition probabilities, they also proposed a new FTC to deal with actuator faults in [27]. When actuator and sensor fault happened simultaneously, Chen et al. [28] proposed two novel fault estimation observers to estimate the faults of MJSs.

In the field of fault estimation and UAV control, there have also been development and research in recent years. A novel robust nonlinear controller is proposed for a UAV system in [29] to achieve Cartesian position trajectory tracking capability. Besides, a novel fault estimation method based on an adaptive observer is designed for taking off mode. Nian et al. [30] designed a robust adaptive fault estimation observer to obtain the actuator fault of a UAV system. Then, they proposed a dynamic output feedback fault-tolerant controller for the stability of the system. For a high-altitude long-endurance UAV, an estimation algorithm is proposed to estimate the three-axis accelerations in [31]. Through flight tests, the algorithm can detect the fault of the accelerometer. Combining with a novel

two-stage Kalman filter, a fault estimation algorithm is designed in [32]. Using a sensor fault detection algorithm and robust Kalman filter, the state parameters of the UAV can be accurately estimated. Goslinski et al. [33] also pay attention to estimation of the state of UAVs, and they proposed a quadrotor model for fault-tolerant observation and a new filtration method.

In order to facilitate analysis and calculation, the theoretical research is mostly based on a linear system. In practical engineering applications, most systems will be affected by nonlinear phenomena, which will destroy the stability of the system. This work will concentrate on the observer-based FTC of MJNSs. We take the specificity of the work environment and the impact of nonlinearity on UAV systems into consideration, and that actuator faults and disturbances are stochastic and hard to predict in advance. Therefore, an observer with adaptive technology is designed in this work. A nonsingular fast terminal sliding mode controller is adopted to improve the rapidity and to avoid the singularity. Moreover, the Lyapunov-Krasovskii functional (LKF) and linear matrix inequalities (LMI) techniques are utilized to guarantee the stochastic stability of the fault-tolerant controller. The main contributions of this paper are summarized as follows:

1. We consider the specificity of the work environment, and the UAV system is regarded as a Markov jump nonlinear system that is proven to be stochastically stable. The nonlinear term is satisfied with the quasi-one-sided Lipschitz condition, which relaxes the constraints and contains more nonlinear information.
2. The FAFE algorithm is utilized to design the adaptive observer to estimate the fault and disturbance, where there is is no need to know the bound of the fault in advance.
3. Based on the estimation given by the observer, a nonsingular fast terminal sliding-mode fault-tolerant controller is applied to control the MJNS, which is proven stable by the LKF.
4. The simulation results on a quadrotor UAV system show the feasibility of the theory.

The rest of this paper is arranged as follows: Section 2 gives a dynamic model of MJNSs and some assumptions, lemmas, and definitions, which will be utilized in the following sections. In Section 3, an adaptive observer is given to estimate the faults and disturbances of the UAV system, and a nonsingular fast terminal sliding-mode fault-tolerant controller whose stability is ensured by the Lyapunov-Krasovskii functional is designed. In Section 4, by the numerical simulation on the UAV system, the feasibility of the method is proven. Eventually, brief conclusions are given in Section 5.

## 2. System Description and Preliminaries

### 2.1. Quadrotor Kinematic Model

To establish the kinematic model of a quadrotor like Figure 1, it is necessary to analyze the motion and the force of the system in the same coordinate system.

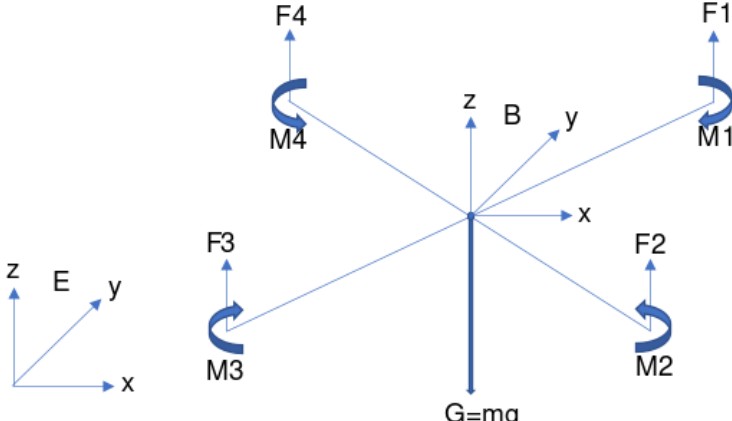

**Figure 1.** Model of a quadrotor.

As is shown in Figure 1, the origin of body coordinate system $B$ is the center of mass of the quadrotor. The x-axis is the roll axis of the body, the y-axis is the pitch axis of the body, and the z-axis is the yaw axis of the body. We select the reference coordinate system $E$, whose origin coincides with the origin of the body coordinate system.

There is a rotation matrix from coordinate system $B$ to $E$, which can be given by the following formula.

$$R_B^E(\phi, \theta, \psi) = R_x(\phi) \cdot R_y(\theta) \cdot R_z(\psi) \tag{1}$$

where,

$$R_x(\phi) = \begin{bmatrix} 1 & 0 & 0 \\ 0 & cos\phi & sin\phi \\ 0 & -sin\phi & cos\phi \end{bmatrix}, \ R_y(\theta) = \begin{bmatrix} cos\theta & 0 & -sin\theta \\ 0 & 1 & 0 \\ sin\theta & 0 & cos\theta \end{bmatrix}, \ R_z(\psi) = \begin{bmatrix} cos\psi & sin\psi & 0 \\ -sin\psi & cos\psi & 0 \\ 0 & 0 & 1 \end{bmatrix} \tag{2}$$

Therefore,

$$R_B^E(\phi, \theta, \psi) = \begin{bmatrix} cos\theta cos\psi & cos\psi sin\theta sin\phi - sin\psi cos\phi & cos\psi sin\theta cos\phi + sin\psi sin\phi \\ cos\theta sin\psi & sin\psi sin\theta sin\phi + cos\psi cos\phi & sin\psi sin\theta cos\phi - cos\psi sin\phi \\ -sin\theta & sin\phi cos\theta & cos\phi cos\theta \end{bmatrix} \tag{3}$$

Regarding the quadrotor system as a rigid body and ignoring the change of the earth's shape and gravitational acceleration, the mass center motion equation can be written as:

$$\begin{cases} \vec{F} = \frac{md\vec{V}}{dt} \\ \vec{M} = \frac{d\vec{H}}{dt} \end{cases} \tag{4}$$

where $V = \begin{bmatrix} v_x & v_y & v_z \end{bmatrix}^T$ is the velocity vector of the center of mass of the quadrotor, $\vec{F}$ is the sum of all external forces acting on the quadrotor, $m$ is the equality of the quadrotor, $\vec{M}$ is the linear momentum, and $\vec{H}$ is moment of momentum of the quadrotor relative to the ground coordinate system.

The elevating force and the torque can be given as:

$$\begin{cases} F = \sum\limits_{i=1}^{4} k_F \omega_i^2 \\ M = \sum\limits_{i=1}^{4} k_M \omega_i^2 \end{cases} \tag{5}$$

where, $k_F$ is the lift coefficient, $k_M$ is the torque quotient, and $\omega_i = \begin{bmatrix} p & q & r \end{bmatrix}^T (i = 1, 2, 3, 4)$ is the angular rate of the ith motor. The conversion relationship between the angular velocity of the Euler angle and the angular velocity of the body is as follows:

$$\begin{bmatrix} p \\ q \\ r \end{bmatrix} = \begin{bmatrix} \dot{\phi} - \dot{\psi}sin\theta \\ \dot{\theta}cos\phi + \dot{\psi}sin\phi cos\theta \\ -\dot{\theta}sin\phi + \dot{\psi}cos\phi cos\theta \end{bmatrix} \tag{6}$$

If the quadrotor is symmetrical, the moment of the inertia matrix can be given as:

$$I_{\phi\theta\psi} = \begin{bmatrix} I_x & 0 & 0 \\ 0 & I_y & 0 \\ 0 & 0 & I_z \end{bmatrix} \tag{7}$$

The equations of motion for angular velocity are as follows:

$$\begin{bmatrix} M_x \\ M_y \\ M_z \end{bmatrix} = \begin{bmatrix} \dot{p}I_x - \dot{r}I_{xz} + qr(I_z - I_y) - pqI_{xz} \\ \dot{q}I_y + pr(I_x - I_z) + (p^2 - r^2)I_{xz} \\ \dot{r}I_z - \dot{p}I_{xz} + pq(I_y - I_x) + qrI_{xz} \end{bmatrix} \tag{8}$$

Therefore, we can obtain the kinematic equations of the quadrotor.

$$
\begin{bmatrix} \ddot{x} \\ \ddot{y} \\ \ddot{z} \\ \ddot{\phi} \\ \ddot{\theta} \\ \ddot{\psi} \end{bmatrix} = \begin{bmatrix} \frac{F_x - K_x \dot{x}}{m} \\ \frac{F_y - K_y \dot{y}}{m} \\ \frac{F_z - mg - K_z \dot{z}}{m} \\ (M_x + (I_x - I_z)qr) / I_x \\ (M_y + (I_z - I_x)qr) / I_y \\ (M_z + (I_x - I_y)qr) / I_z \end{bmatrix} \tag{9}
$$

Design the input of the system,

$$
\begin{cases} U_1 = k_F l \left( \omega_1^2 - \omega_2^2 - \omega_3^2 + \omega_4^2 \right) \\ U_2 = k_F l \left( \omega_1^2 + \omega_2^2 - \omega_3^2 + \omega_4^2 \right) \\ U_3 = k_M \left( \omega_1^2 - \omega_2^2 + \omega_3^2 - \omega_4^2 \right) \\ U_4 = k_f \left( \omega_1^2 + \omega_2^2 + \omega_3^2 + \omega_4^2 \right) \end{cases} \tag{10}
$$

where $U_1$, $U_2$, $U_3$ represent the control input of roll, pitch, and yaw, respectively, and $U_4$ is the input control of height.

Considering the UAV flying at low speed, we can simplify the model:

$$
\begin{bmatrix} \ddot{x} \\ \ddot{y} \\ \ddot{z} \\ \ddot{\phi} \\ \ddot{\theta} \\ \ddot{\psi} \end{bmatrix} = \begin{bmatrix} \frac{(sin\phi sin\psi + cos\phi sin\theta cos\psi)U_4}{m} \\ \frac{(-sin\phi cos\psi + cos\phi sin\theta sin\psi)U_4}{m} \\ \frac{(sin\theta cos\phi)U_4}{m} \\ \frac{U_1 + \dot{\theta}\dot{\psi}\left( I_y - I_z \right)}{I_x} \\ \frac{U_2 + \dot{\phi}\dot{\psi}\left( I_z - I_x \right)}{I_y} \\ \frac{U_3 + \dot{\phi}\dot{\theta}\left( I_x - I_y \right)}{I_z} \end{bmatrix} \tag{11}
$$

In this paper, we pay more attention to the attitude of the quadrotor, and complete further simplification of the model [34]:

$$
\begin{bmatrix} \dot{\phi} \\ \dot{\theta} \\ \dot{\psi} \\ \ddot{\phi} \\ \ddot{\theta} \\ \ddot{\psi} \end{bmatrix} = \begin{bmatrix} p \\ q \\ r \\ \frac{U_1}{I_x} \\ \frac{U_2}{I_y} \\ \frac{U_3}{I_z} \end{bmatrix} \tag{12}
$$

Selecting the equilibrium points as [34], the state space equation can be written into:

$$
\begin{cases} \dot{x} = Ax + Bu \\ y = Cx \end{cases} \tag{13}
$$

where $A = \begin{bmatrix} O_{3\times3} & I_{3\times3} \\ O_{3\times3} & O_{3\times3} \end{bmatrix}$, $B = \begin{bmatrix} O_{3\times3} \\ I_{\phi\theta\psi}^{-1} \end{bmatrix}$, $C = \begin{bmatrix} I_{3\times3} & O_{3\times3} \end{bmatrix}$.

### 2.2. Markov Jump Nonlinear Systems Dynamic Model

In practice, under the interference of external factors, actuator failure, uncertain disturbance, and other influencing factors easily appear. Therefore, this paper comprehensively considers these factors and gives the following dynamic model of a Markov jump nonlinear system.

$$
\begin{cases} \dot{x}(t) = A(r(t))x(t) + B(r(t))(u(t) + f(x(t), t)) + F_a(r(t))f_a(t) + B_\omega(r(t))\omega(t) \\ y(t) = C(r(t))x(t) \end{cases} \tag{14}
$$

where, $x(t) \in \mathbb{R}^n$ is the state vector, $u(t) \in \mathbb{R}^m$ is the control input vector, and $y(t) \in \mathbb{R}^p$ is the measured output vector. The coefficient matrices $A(r(t)) \in \mathbb{R}^{n \times n}$, $B(r(t)) \in \mathbb{R}^{n \times m}$, $C(r(t)) \in \mathbb{R}^{p \times n}$ are known constant matrices. $f(x(t), t)$ represents the nonlinear function, $f_a(t)$ represents actuator failure, and $\omega(t)$ is the disturbance. In addition, $F_a(r(t))$ and $B_\omega(r(t))$ are the constant matrices with appropriate dimensions and column full rank.

Let $\{r(t), t \geq 0\}$ be a Markov process with right continuous trajectories on the probability space $(\Omega, F, P)$. $r(t)$ takes values in the finite set $\mathcal{N} = \{1, 2, \cdots, n\}$.

The state transition matrix $\Pi = (\pi_{ij})$ is set as:

$$P_r\{r(t + \delta) = j | r(t) = i\} = \begin{cases} \pi_{ij}\delta + o(\delta), & i \neq j \\ 1 + \pi_{ii}\delta + o(\delta), & i = j \end{cases} \tag{15}$$

where, $\delta > 0$, $\lim\limits_{\delta \to 0} \frac{o(\delta)}{\delta} = 0$. When $i \neq j$, $\pi_{ij} > 0$, which is the transition Rates (TRs) from state $i$ at time $t$ to state $j$ at time $t + \delta$ and satisfies that $\pi_{ii} = -\sum\limits_{j=1, i \neq j}^{n} \pi_{ij} < 0$.

Therefore, the transition rate matrix can be expressed as:

$$\Pi = \begin{pmatrix} \pi_{11} & \cdots & \pi_{1n} \\ \vdots & \ddots & \vdots \\ \pi_{n1} & \cdots & \pi_{nn} \end{pmatrix} \tag{16}$$

When the system is in the state $i$, that is, when $r(t) = i$, $i \in \mathcal{N}$, $A(r(t))$, $B(r(t))$, $C(r(t))$ can be simplified to the real constant matrix $A_i$, $B_i$, $C_i$, and the nonlinear function $f(x(t), t)$ is written as $f_i(x(t), t)$.

System (14) can be rewritten as follows:

$$\begin{cases} \dot{x}(t) = A_i x(t) + B_i(u(t) + f_i(x(t), t)) + F_{ai}f_{ai}(t) + B_{\omega i}\omega(t) \\ y(t) = C_i x(t) \end{cases} \tag{17}$$

**Remark 1.** *In this paper, $\mathcal{N}$ is the set of positive integers, and $\mathbb{R}$ is the set of real numbers. $\mathbb{R}^n$ denotes the n-dimensional vector space and $\mathbb{R}^{m \times n}$ denotes the space of all $m \times n$-dimensional matrices. $A^T$ represents the transpose of matrix $A$. $\|\cdot\|$ indicates the Euclidean norm of the vector. The inner product of vectors $x, y \in \mathbb{R}^m$ is denoted by $\langle x, y \rangle$, and $\langle x, y \rangle = x^T y$.*

**Assumption 1.** *The nonlinear term $f_i(x(t), t)$ is continuous with $x(t)$ and satisfies the following inequality on the domain of definition:*

$$\langle f_i(x(t), t) - f_i(\hat{x}(t), t), x(t) - \hat{x}(t) \rangle \leq (x - \hat{x})^T M(x - \hat{x}), \ x(t) \in \mathbb{R}^n \tag{18}$$

*where, $M$ is a real symmetric matrix and the one-sided Lipschitz constant matrix, and $\hat{x}$ is the estimation of $x$.*

**Assumption 2** ([35]). *The nonlinear term $f_i(x(t), t)$ satisfies the quadratic inner-boundedness condition. If $\exists \alpha, \beta \epsilon R$ is in a continuous closed region containing the origin, such that $\forall x, \hat{x} \in \mathbb{R}^n$,*

$$\|f_i(x(t), t) - f_i(\hat{x}(t), t)\|^2 \leq \alpha \|x(t) - \hat{x}(t)\|^2 + \beta \langle x(t) - \hat{x}(t), f_i(x(t), t) - f_i(\hat{x}(t), t) \rangle \tag{19}$$

**Remark 2.** *The two assumptions are used to describe the quasi-one-sided Lipschitz condition. In the system description, the nonlinear terms are assumed to satisfy this condition. Compared with traditional Lipschitz, this quasi-one-sided Lipschitz condition relaxes restrictions, which can facilitate the calculation of LMI.*

**Definition 1** ([36])**.** *For any $e_0 \in \mathbb{R}^n$, $r_0 \in \mathcal{N}$; when $u(t) = 0$, if the MJSs satisfy the following inequality, the MJSs are stochastically stable.*

$$E\left\{\int_0^t \|e(t)\|^2 dt \middle| e_0, r_0\right\} < \infty \tag{20}$$

**Definition 2** ([37])**.** *For the stochastic Lyapunov-Krasovskii function $V(x(t), i)$, the weak infinitesimal operator $\mathcal{L}$ meets with*

$$\begin{aligned}
\mathcal{L}V(x(t), i) &= \lim_{\Delta \to 0^+} \frac{1}{\Delta}[E\{V(x(t+\Delta), r_{t+\Delta})|x(t), r_t = i\} - V(x(t), i)] \\
&= V_t(x(t), i) + V_x(x(t), i)\dot{x}(t) + \sum_{j=1}^n \pi_{ij} V(x(t), j)
\end{aligned} \tag{21}$$

**Lemma 1** ([38])**.** *For given real matrices $\mathcal{A}$, $\mathcal{B}$, $\mathcal{C}$, $\mathcal{D}$, $\mathcal{E}$, $\mathcal{F}$, and $\Xi$, there exists a symmetrical matrix $P > 0$, such that*

$$\begin{bmatrix} P\mathcal{A}^T + \mathcal{A}P + \Xi & \mathcal{B} & P\mathcal{C}^T \\ * & \mathcal{D} & \mathcal{E}^T \\ * & * & \mathcal{F} \end{bmatrix} < 0 \tag{22}$$

**Lemma 2** ([39])**.** *According to the classical Paul and Peter inequality, for a scalar $\rho > 0$, real vectors $x$ and $y$, and a symmetric positive definite matrix $R$, only for real Euclidean space $\mathbb{E}$, endowed with the scalar product $\langle \cdot | \cdot \rangle$, the following inequality holds:*

$$2x^T y \le \frac{1}{\rho} x^T R x + \rho y^T R^{-1} y \tag{23}$$

**Lemma 3** ([40])**.** *Consider a nonsingular fast terminal sliding mode surface:*

$$s = z_1 + \kappa_{i1}^{-1} z_1^\gamma + \kappa_{i2}^{-1} z_2^{\frac{p}{q}} \tag{24}$$

*If $s(0) \ne 0$, the convergence time to $s = 0$ can be given as:*

$$\begin{aligned}
T &= \int_0^{|z_1(0)|} \frac{k_2^{q/p}}{(z_1(t) + k_1 z_1)^{q/p}} dz_1 \\
&= \frac{\frac{p}{q}|z_1(0)|^{1-q/p}}{k_1\left(\frac{p}{q} - 1\right)} F\left(\frac{q}{p}, \frac{\frac{p}{q} - 1}{(\lambda - 1)\frac{p}{q}}; 1 + \frac{\frac{p}{q} - 1}{(\lambda - 1)\frac{p}{q}}; -k_1|z_1(0)|^{\lambda-1}\right)
\end{aligned} \tag{25}$$

## 3. Main Results

### 3.1. Observer Design and Fault Estimation

Considering the fault and disturbance, this paper sets that $F_i g(t) = F_{ai} f_a(t) + B_{\omega i} \omega(t)$. Then, the following adaptive observer is designed for the dynamic system model (17):

$$\begin{cases} \dot{\hat{x}}(t) = A_i \hat{x}(t) + B_i(u(t) + f(\hat{x}(t), t)) + F_i \hat{g}(t) + L_i(y(t) - \hat{y}(t)) \\ \hat{y}(t) = C_i \hat{x}(t) \end{cases} \tag{26}$$

where, $L_i$ is the gain matrix of the observer, $\hat{x}(t)$ is the estimation of the true state $x(t)$, $\hat{y}(t)$ is the estimation of the true output $y(t)$, and $\hat{g}(t)$ is the estimation of $g(t)$.

Define the estimation error as: $e(t) = x(t) - \hat{x}(t)$, $e_y(t) = y(t) - \hat{y}(t)$, $e_a(t) = g(t) - \hat{g}(t)$. Then, the error equation is given by $\dot{e}(t) = \dot{x}(t) - \dot{\hat{x}}(t)$.

Considering Equations (17) and (26), the expression of the error dynamic system is:

$$\dot{e}(t) = (A_i - L_i C_i)e(t) + B_i e_f(x(t), \hat{x}(t)) + F_i e_g(t) \tag{27}$$

A fast adaptive fault estimation (FAFE) algorithm can be adopted in the adaptive observers. The estimation of the derivatives of faults can be written as:

$$\dot{\hat{g}} = -\Lambda T_i\big(e_y + \zeta \dot{e}_y\big) \tag{28}$$

where $\zeta > 0$ is a scalar, $\Lambda \in \mathbb{R}^{n \times p}$ is a symmetric positive definite matrix, $T_i \in \mathbb{R}^{r \times p}$, and their specific definitions will be given in Theorem 1.

From Equation (28), the uniformly ultimate boundedness of $e_x(t)$ and $e_y(t)$ can be achieved.

*3.2. Observer-Based Nonsingular Fast Terminal Sliding Mode Fault-Tolerant Control Design*

This section designs a nonsingular fast terminal sliding mode fault-tolerant controller (NFTSM-FTC) for the system (14). Considering Equation (17), the controller can be designed as:

$$\begin{cases} \dot{x}_1 = x_2 \\ \dot{x}_2 = A_i x_2 + B_i(u(t) + f_i(x_2(t), t)) + F_i \hat{g}(t) + F_i e_g(t) \end{cases} \tag{29}$$

where $F_i \hat{g}(t)$ represents the estimation of fault and disturbance that is known, $e_g(t)$ includes the uncertain part of fault and disturbance, and $F_i e_g(t) = F_{ai} e_{ai}(t) + B_{\omega i} e_\omega(t)$.

The tracking error $z_1$ and the second error $z_2$ are defined as follows:

$$\begin{aligned} z_1 &= x_1 - x_d \\ z_2 &= x_2 - \varpi \end{aligned} \tag{30}$$

where $\varpi = -\chi z_1 + \dot{x}_d$ is a virtual control.

Then, by calculating the differential of the equation, we can get:

$$\begin{aligned} \dot{z}_1 &= \dot{x}_1 - \dot{x}_d \\ \dot{z}_2 &= \dot{x}_2 - \dot{\varpi} \end{aligned} \tag{31}$$

Considering the system (29), we design the following sliding surface for each Markov mode $i \in \mathcal{N}$:

$$s = z_1 + \kappa_{i1}^{-1} z_1^\gamma + \kappa_{i2}^{-1} z_2^{\frac{p}{q}} \tag{32}$$

where $s = (s_1, \cdots, s_n)^T \in \mathbb{R}^n$ is the sliding variable, $\kappa_{i1}^{-1} = diag\big\{\kappa_{11}^{-1}, \ldots, \kappa_{1n}^{-1}\big\}$ and $\kappa_2^{-1} = diag\big\{\kappa_{21}^{-1}, \ldots, \kappa_{2n}^{-1}\big\}$ are diagonal positive definite matrices, $p, q > 0$, and they are odd numbers satisfying the relation $1 < p/q < 2$, $\gamma > p/q$.

Calculating $\dot{s}$, we can get:

$$\dot{s} = \dot{z}_1 + \kappa_{i1}^{-1} z_1^{\gamma-1} \cdot \dot{z}_1 + \kappa_{i2}^{-1} \frac{p}{q} z_2^{\frac{p}{q}-1} \cdot \dot{z}_2 \tag{33}$$

where $z_1^{\gamma-1} = diag\big\{|z_{11}|^{\gamma-1} sgn(z_{11}), \ldots, |z_{n1}|^{\gamma-1} sgn(z_{n1})\big\}$,
$z_2^{\frac{p}{q}-1} = diag\big\{|z_{21}|^{\frac{p}{q}-1} sgn(z_{21}), \ldots, |z_{2n}|^{\frac{p}{q}-1} sgn(z_{2n})\big\}$.

Design the following NFTSM-FTC:

$$u(t) = u_1 + u_2 + u_3 \tag{34}$$

where

$$u_1 = -B_i^+ (A_i x_2 + F_i \hat{g}(t) - \dot{\kappa}) - f_i(x_2(t), t) \tag{35}$$

$$u_2 = -\frac{q}{p} B_i^+ \kappa_{i2} z_2^{1-\frac{p}{q}} \cdot \big(\dot{z}_1 + \kappa_{i1}^{-1} z_1^{\gamma-1} \cdot \dot{z}_1\big) \tag{36}$$

$$u_3 = -\frac{q}{p} B_i^+ \kappa_{i2} \cdot z_2^{1-\frac{p}{q}} \varepsilon |s_1| sgn(s) \tag{37}$$

where $B_i^+$ satisfying the relation $B_i B_i^+ = I$, is the generalized inverse matrix of $B_i$, $\varepsilon \in \mathcal{R}^+$.

**Proof.** Choose the Lyapunov function as:

$$V_1(x(t), i) = \frac{1}{2} s^T s \tag{38}$$

According to the Definition 2 and paper [14], we have

$$
\begin{aligned}
\mathcal{L}V_1(x(t), i) &= \frac{1}{2} \lim_{\Delta \to 0} \frac{1}{\Delta} \left\{ \sum_{j=1, i \neq j}^{\mathcal{N}} Pr\{r(t+\delta) = j \mid r(t) = i\} s^T(t+\Delta) s(t+\Delta) \right. \\
&\quad \left. + Pr\{r(t+h) = i \mid r(t) = i\} s^T(t+\Delta) s(t+\Delta) - s^T(t) s(t) \right\} \\
&= \frac{1}{2} \lim_{\Delta \to 0} \frac{1}{\Delta} \left\{ \sum_{j=1, j \neq i}^{\mathcal{N}} \frac{\varrho_{ij}(F_{Xi}(\delta + \Delta) - F_{Xi}(\delta))}{1 - F_{Xi}(h)} s^T(t+\Delta) s(t+\Delta) \right. \\
&\quad \left. + \frac{1 - F_{Xi}(\delta + \Delta)}{1 - F_{Xi}(\delta)} s^T(t+\Delta) s(t+\Delta) - s^T(t) s(t) \right\} \\
&= s^T(t) \dot{s}(t) + \frac{1}{2} \sum_{j=1}^{\mathcal{N}} \pi_{ij} \\
&= s^T(t) \dot{s}(t)
\end{aligned}
\tag{39}
$$

where $\delta$ is the time from mode $i$ to mode $j$; $\varrho$ represents the probability from mode $i$ to mode $j$. $F_{Xi}(x)$ denotes the cumulative distribution function (CDF) of $x$ on mode $i$.

Therefore,

$$
\begin{aligned}
\mathcal{L}V_1(x(t), i) &= s^T \dot{s} \\
&= s^T \left( \dot{z}_1 + \kappa_{i1}^{-1} z_1^{\gamma-1} \cdot \dot{z}_1 + \kappa_{i2}^{-1} \frac{p}{q} z_2^{\frac{p}{q}-1} \cdot \dot{z}_2 \right) \\
&= s^T \left( \dot{z}_1 + \kappa_{i1}^{-1} z_1^{\gamma-1} \cdot \dot{z}_1 \right) \\
&\quad + s^T \kappa_{i2}^{-1} \frac{p}{q} z_2^{\frac{p}{q}-1} \cdot \left( A_i x_2 + B_i(u(t) + f_i(x_2(t), t)) + F_i \hat{g}(t) + F_i e_g(t) - \dot{\omega} \right)
\end{aligned}
\tag{40}
$$

Substituting (34) into (40), it becomes:

$$
\begin{aligned}
\mathcal{L}V_1(x(t), i) &\leq s^T \kappa_{i2}^{-1} \frac{p}{q} z_2^{\frac{p}{q}-1} \cdot \left( F_i e_g(t) \right) - s^T \varepsilon |s| sgn(s) \\
&\leq \sum_{j=1}^{n} \left\{ \left( \kappa_{i2}^{-1} \frac{p}{q} z_2^{\frac{p}{q}-1} \right)_j \cdot |s_j| \cdot \| F_i e_g(t) \|_\infty \right\} - s^T \varepsilon |s| sgn(s) \\
&\leq -\varepsilon \|s\|^2
\end{aligned}
\tag{41}
$$

Since $\varepsilon > 0$, when $s \neq 0$, we can get $\mathcal{L}V_1(x(t), i) < 0$. According to Lemma 3, the time in which the system state reaches the equilibrium point is finite, and the convergence time $T$ can be given. This is if and only if $s = 0$, $\mathcal{L}V_1(x(t), i) = 0$. $\square$

**Theorem 1.** *The error dynamic system (27) and the control system (29) are stochastically stable, if there exist symmetrical matrices $Q_i > 0$, $Q > 0$, $O_i > 0$, $O > 0$, $\Lambda > 0$, $M_i > 0$, which is the Lipschitz constant matrix, and $O_i \in \mathbb{R}^{r \times r}$, $T_i \in \mathbb{R}^{r \times p}$, such that the following conditions hold*

$$\sum_{j=1}^{n} \pi_{ij}(-O_j) \leq -O, \ \sum_{j=1}^{n} \pi_{ij} Q_j \leq Q$$

$$F_i^T Q_i = T_i C_i$$

$$\Phi = \begin{bmatrix} \Phi_{11} & \Phi_{12} & \Phi_{13} & 0 \\ * & \Phi_{22} & \Phi_{23} & 0 \\ * & * & \Phi_{33} & 0 \\ * & * & * & \Phi_{44} \end{bmatrix} < 0$$

(42)

*where*

$\Phi_{11} = Q_i(A_i - L_i C_i) + (A_i - L_i C_i)^T Q_i + \sum_{j=1}^{n} \pi_{ij} Q_j + k_i M_i + \alpha_i m_i$

$\Phi_{12} = -\frac{1}{\zeta}\left(A_i^T Q_i F_i - (Q_i L_i C_i)^T F_i\right)$

$\Phi_{13} = Q_i B_i + \frac{1}{2}(\beta_i m_i - k_i)I$

$\Phi_{22} = -\frac{2}{\zeta} F_i^T Q_i F_i + \frac{1}{\zeta\rho} O_i + \frac{1}{\zeta} \sum_{j=1}^{n} \pi_{ij}\Lambda^{-1}$

$\Phi_{23} = -\frac{1}{\zeta} F_i^T Q_i B_i$

$\Phi_{33} = -m_i I$

$\Phi_{44} = -\frac{\rho}{\zeta}(O + \Lambda^{-1} O_i \Lambda^{-1})$

**Proof.** Choose the Lyapunov function as:

$$V_2(x(t), i) = e(t)^T Q_i e(t) + \frac{1}{\zeta} e_g^T(t)\Lambda^{-1} e_g - \frac{\rho}{\zeta} g^T(t)\Lambda^{-1} O_i \Lambda^{-1} g(t)$$

(43)

Like the proof of the Lyapunov function (39), we now calculate $\mathcal{L}V_2(x(t), i)$,

$$\begin{aligned}
\mathcal{L}V_2(x(t), i) =\ & e(t)^T\left(Q_i(A_i - L_i C_i) + (A_i - L_i C_i)^T Q_i\right)e(t) \\
& + 2e(t)^T Q_i B_i e_f(x(t), \hat{x}(t)) + 2e(t)^T F_i^T Q_i e_g(t) + \frac{2}{\zeta} e_g^T(t)\Lambda^{-1}\dot{e}_g(t) \\
& - \frac{\rho}{\zeta}\dot{g}^T(t)\Lambda^{-1} O_i \Lambda^{-1} g(t) - \frac{\rho}{\zeta} g^T(t)\Lambda^{-1} O_i \Lambda^{-1}\dot{g}(t) + \sum_{j=1}^{n} \pi_{ij} V_2(x(t), i) \\
=\ & e(t)^T\left(Q_i(A_i - L_i C_i) + (A_i - L_i C_i)^T Q_i\right)e(t) + 2e(t)^T Q_i B_i e_f(x(t), \hat{x}(t)) \\
& - \frac{2}{\zeta} e_g^T(t) T_i C_i (A - LC) e_x(t) - \frac{2}{\zeta} e_g^T(t) F_i^T Q_i\left(F_i e_g(t) + B_i e_f(x(t), \hat{x}(t))\right) \\
& - \frac{2}{\zeta} e_g^T(t)\Lambda^{-1}\dot{g}(t) - \frac{\rho}{\zeta}\left(\dot{g}^T(t) + g^T(t)\right)\Lambda^{-1} O_i \Lambda^{-1}(\dot{g}(t) + g(t)) + \sum_{j=1}^{n} \pi_{ij} V_2(x(t), i)
\end{aligned}$$

(44)

According to Lemma 2,

$$-\frac{2}{\zeta} e_g^T(t)\Lambda^{-1}\dot{g}(t) \leq \frac{1}{\zeta\rho} e_g^T(t) O_i e_g(t) + \frac{\rho}{\zeta}\dot{g}^T(t)\Lambda^{-1} O_i \Lambda^{-1}\dot{g}(t)$$

(45)

Substitute (45) into (44),

$$
\begin{aligned}
\mathcal{L}V_2(x(t),i) \leq\ & e(t)^T\Big(Q_i(A_i - L_iC_i) + (A_i - L_iC_i)^T Q_i\Big)e(t) + 2e(t)^T Q_i B_i\, e_f(x(t),\hat{x}(t)) \\
& - \frac{2}{\zeta}e_g^T(t)T_iC_i(A - LC)e_x(t) - \frac{2}{\zeta}e_g^T(t)F_i^T Q_i\Big(F_i e_g(t) + B_i e_f(x(t),\hat{x}(t))\Big) \\
& + \frac{1}{\zeta\rho}e_g^T(t)O_i e_g(t) + \frac{\rho}{\zeta}\dot{g}^T(t)\Lambda^{-1}O_i\Lambda^{-1}\dot{g}(t) \\
& - \frac{\rho}{\zeta}\Big(\dot{g}^T(t) + g^T(t)\Big)\Lambda^{-1}O_i\Lambda^{-1}(\dot{g}(t) + g(t)) + \sum_{j=1}^{n}\pi_{ij}V_2(x(t),i)
\end{aligned}
\tag{46}
$$

Considering the assumptions (1) and (2), the following equations can be given:

$$
\begin{cases}
k_i e(t)^T M_i e(t) - k_i e_{fi}(x(t),\hat{x}(t))^T e(t) \geq 0 \\
\alpha_i m_i e(t)^T e(t) - m_i e_{fi}(x(t),\hat{x}(t))e_{fi}(x(t),\hat{x}(t))^T + \beta_i m_i e(t)^T e_{fi}(x(t),\hat{x}(t)) \geq 0
\end{cases}
\tag{47}
$$

where $k_i, m_i \in \mathcal{R}^+$.

Add Formula (47) to the right of Formula (46), and then we can get:

$$
\begin{aligned}
\mathcal{L}V_2(x(t),i) \leq\ & e(t)^T\Big(Q_i(A_i - L_iC_i) + (A_i - L_iC_i)^T Q_i\Big)e(t) + 2e(t)^T Q_i B_i\, e_f(x(t),\hat{x}(t)) \\
& - \frac{2}{\zeta}e_g^T(t)T_iC_i(A - LC)e_x(t) - \frac{2}{\zeta}e_g^T(t)F_i^T Q_i\Big(F_i e_g(t) + B_i e_f(x(t),\hat{x}(t))\Big) \\
& + \frac{1}{\zeta\rho}e_g^T(t)O_i e_g(t) + \frac{\rho}{\zeta}\dot{g}^T(t)\Lambda^{-1}O_i\Lambda^{-1}\dot{g}(t) \\
& - \frac{\rho}{\zeta}\Big(\dot{g}^T(t) + g^T(t)\Big)\Lambda^{-1}O_i\Lambda^{-1}(\dot{g}(t) + g(t)) + k_i e(t)^T M_i e(t) \\
& - k_i e_{fi}(x(t),\hat{x}(t))^T e(t) + \alpha_i m_i e(t)^T e(t) - m_i e_{fi}(x(t),\hat{x}(t))e_{fi}(x(t),\hat{x}(t))^T \\
& + \beta_i m_i e(t)^T e_{fi}(x(t),\hat{x}(t)) \\
& + e(t)^T\sum_{j=1}^{n}\pi_{ij}Q_j e(t) + \frac{1}{\zeta}e_g^T(t)\sum_{j=1}^{n}\pi_{ij}\Lambda^{-1}e_g(t) - \frac{\rho}{\zeta}g^T(t)Og(t)
\end{aligned}
\tag{48}
$$

The inequation (48) can be written as:

$$
\mathcal{L}V_2(x(t),i) <
\begin{bmatrix}
e(t) \\
e_g(t) \\
e_{fi}(x(t),\hat{x}(t)) \\
g(t)
\end{bmatrix}^T
\begin{bmatrix}
\mathcal{A}_i & \mathcal{B}_i & \mathcal{C}_i & 0 \\
* & \mathcal{D}_i & \mathcal{E}_i & 0 \\
* & * & \mathcal{F}_i & 0 \\
* & * & * & \mathcal{G}_i
\end{bmatrix}
\begin{bmatrix}
e(t) \\
e_g(t) \\
e_{fi}(x(t),\hat{x}(t)) \\
g(t)
\end{bmatrix}
\tag{49}
$$

where

$\mathcal{A}_i = Q_i(A_i - L_iC_i) + (A_i - L_iC_i)^T Q_i + \sum\limits_{j=1}^{n}\pi_{ij}Q_j + k_i M_i + \alpha_i m_i$

$\mathcal{B}_i = -\frac{1}{\zeta}\Big(A_i^T Q_i F_i - (Q_i L_i C_i)^T F_i\Big)$

$\mathcal{C}_i = Q_i B_i + \frac{1}{2}(\beta_i m_i - k_i)I$

$\mathcal{D}_i = -\frac{2}{\zeta}F_i^T Q_i F_i + \frac{1}{\zeta\rho}O_i + \frac{1}{\zeta}\sum\limits_{j=1}^{n}\pi_{ij}\Lambda^{-1}$

$\mathcal{E}_i = -\frac{1}{\zeta}F_i^T Q_i B_i$

$\mathcal{F}_i = -m_i I$

$\mathcal{G}_i = -\frac{\rho}{\zeta}(O + \Lambda^{-1}O_i\Lambda^{-1})$

Based on Lemma 1, we obtain that

$$
\begin{aligned}
\mathcal{L}V_2(x(t),i) &\leq \eta^T(t)\Phi_i\eta(t) \\
&= -\eta^T(t)(-\Phi_i)\eta(t) \\
&\leq -\mu\|\eta(t)\|^2
\end{aligned} \tag{50}
$$

where $\eta^T(t) = \begin{bmatrix} e^T(t) & e_g^T(t) & e_f^T(x(t),\hat{x}(t)) & g(t) \end{bmatrix}$, $\Phi_i = \begin{bmatrix} \mathcal{A}_i & \mathcal{B}_i & \mathcal{C}_i & 0 \\ * & \mathcal{D}_i & \mathcal{E}_i & 0 \\ * & * & \mathcal{F}_i & 0 \\ * & * & * & \mathcal{G}_i \end{bmatrix}$, and

$\mu = \min\limits_{i\in\mathcal{N}}\{\sigma(-\Phi_i)\} > 0$.

According to the Dynkin theorem,

$$
E\{V(x(t),i)\} - E\{V_0\} \leq -\mu E\left\{\int_0^t (\mathcal{L}V(x(t),i))ds\right\}
$$

$$
E\left\{\int_0^t (\mathcal{L}V(x(t),i))ds\right\} \leq -\frac{1}{\mu}[E\{V(x(t),i)\} - E\{V_0\}] \leq \frac{1}{\mu}E\{V_0\} \tag{51}
$$

$$
E\left\{\int_0^t \|e(t)\|^2 dt \Big| e_0, r_0\right\} < \infty
$$

Based on Definition 1, the system is stochastically stable. □

## 4. Simulation Study

In this paper, the quadrotor of Canada company is used as the simulation object of the fault-tolerant algorithm [41]. The quadrotor system used in the experiment is presented in Figures 2–4. This quadrotor system can upload control programs and faults to the onboard processor via Wi-Fi, which is helpful for fault injection and experiments. This experimental platform includes a Qdrone quadrotor, an OptiTrack motion capture system, and a ground station with a PC and a router. Through the data from 12 OptiTrack Flex-3 motion capture cameras, the OptiTrack Tools software on the PC can give the real-time flight status of the Qdrone quadrotor, so as to realize online control of the quadrotor. In addition, faults, nonlinear factors, and disturbances can be designed on the PC for experiments.

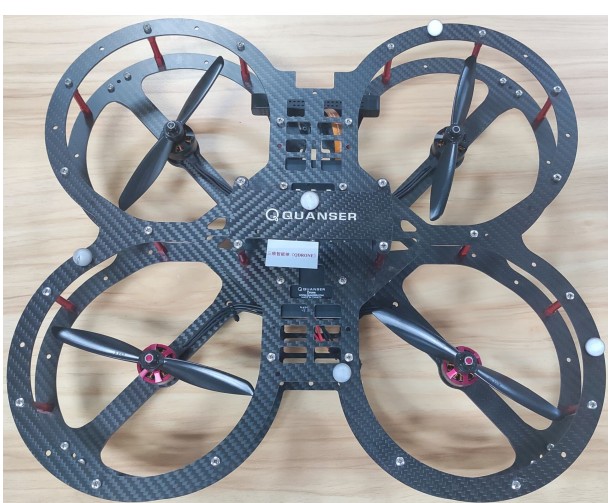

**Figure 2.** Qdrone.

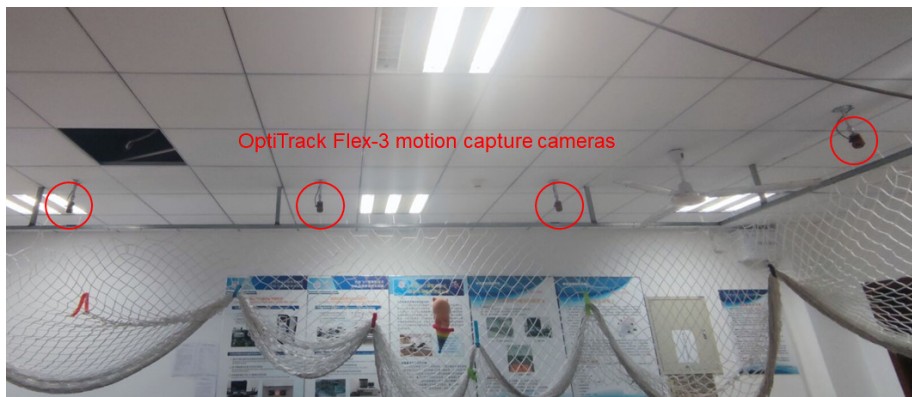

**Figure 3.** OptiTrack Flex-3 motion capture cameras.

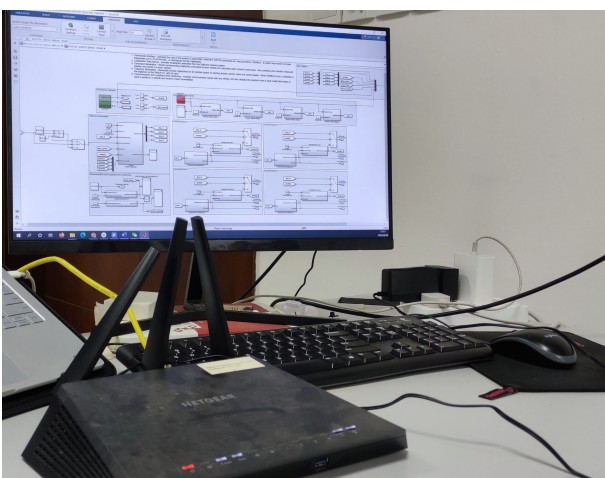

**Figure 4.** The ground station.

According to [41], we choose the appropriate parameters of the quadrotor UAV system in Table 1.

**Table 1.** Parameters of the Qdrone.

| Parameter | Numerical Value | Unit |
|:---:|:---:|:---:|
| $m$ | 1.121 | kg |
| $g$ | 9.80 | $m/s^2$ |
| $I_x$ | 0.010 | $kg \cdot m^2$ |
| $I_y$ | 0.008 | $kg \cdot m^2$ |
| $I_z$ | 0.015 | $kg \cdot m^2$ |

We choose $N = 4$, and the transition rates matrix is chosen as:

$$\Pi = \begin{bmatrix} -0.36 & 0.16 & 0.16 & 0.04 \\ 0.64 & -0.84 & 0.16 & 0.04 \\ 0.64 & 0.16 & -0.84 & 0.04 \\ 0.64 & 0.16 & 0.16 & -0.96 \end{bmatrix} \tag{52}$$

The trajectories of the system mode are presented in Figure 5.

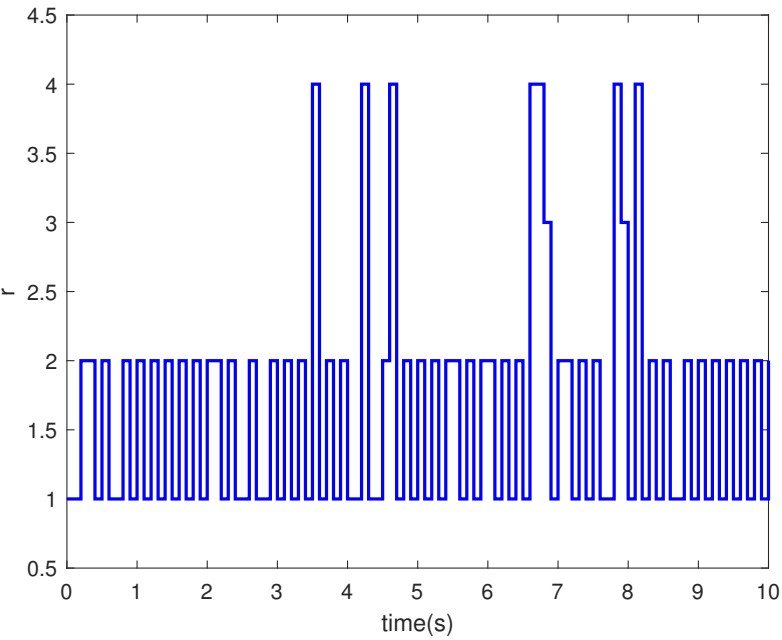

**Figure 5.** System mode.

When some actuators considering the rolling and pitching directions fail, the matrix B will change. Four different modes of matrix $B$ are given as follows:

$$B_1 = \begin{bmatrix} 0 & 0 & 0 \\ 0 & 0 & 0 \\ 0 & 0 & 0 \\ 100 & 0 & 0 \\ 0 & 125 & 0 \\ 0 & 0 & 67 \end{bmatrix}, B_2 = \begin{bmatrix} 0 & 0 & 0 \\ 0 & 0 & 0 \\ 0 & 0 & 0 \\ 75 & 0 & 0 \\ 0 & 125 & 0 \\ 0 & 0 & 67 \end{bmatrix}, B_3 = \begin{bmatrix} 0 & 0 & 0 \\ 0 & 0 & 0 \\ 0 & 0 & 0 \\ 100 & 0 & 0 \\ 0 & 93.75 & 0 \\ 0 & 0 & 67 \end{bmatrix}, B_4 = \begin{bmatrix} 0 & 0 & 0 \\ 0 & 0 & 0 \\ 0 & 0 & 0 \\ 75 & 0 & 0 \\ 0 & 93.75 & 0 \\ 0 & 0 & 67 \end{bmatrix} \quad (53)$$

where $B_1$ represents the healthy system and others represent actuator faults of different direction. $B_2$ represents some actuator failure in the roll direction, $B_3$ represents some actuator failure in the pitch direction, and $B_4$ represents actuator failure in both the roll direction and pitch direction.

Taking the parameters $p = 7$, $q = 5$, $\gamma = 2$, $\varepsilon = 0.1$, $\zeta = 1$, $m = 0.5$, $k = 0.5$. The initial attitude of the UAV system is shown as $\begin{bmatrix} \phi & \theta & \psi \end{bmatrix}^T = \begin{bmatrix} 0.2 & -0.2 & 0.5 \end{bmatrix}^T$, and the initial angle rate is shown as $\begin{bmatrix} \dot\phi & \dot\theta & \dot\psi \end{bmatrix}^T = \begin{bmatrix} 0 & 0 & 0 \end{bmatrix}^T$. Besides, the nonlinear term is $f(x(t), t) = 5sin(\pi t)$. Moreover, the actuator faults and disturbances are given as:

$$\omega(t) = \begin{cases} 0, 0 \leq t < 2 \\ e^{-0.8t}, 2 \leq t \leq 10 \end{cases} \quad (54)$$

$$f_a(t) = \begin{cases} 0, 0 \leq t < 0.5 \\ te^{-0.8t}, 0.5 \leq t < 2 \\ 0.8sin(\pi t), 2 \leq t \leq 10 \end{cases} \quad (55)$$

By solving the linear matrix inequalities in Theorem 1, the parameter matrices can be given as follows:

$$Q_1 = \begin{bmatrix} 76.4090 & -3.0527 & -7.5335 & -0.5193 & -0.0131 & 0.0455 \\ -3.0527 & 64.9062 & -6.5765 & 0.0013 & -0.4108 & 0.0449 \\ -7.5335 & -6.5765 & 97.3179 & 0.0091 & -0.0083 & -0.7604 \\ -0.5193 & 0.0013 & 0.0091 & 0.0072 & 0.0021 & 0.0021 \\ -0.0131 & -0.4108 & -0.0083 & 0.0021 & 0.0049 & 0.0022 \\ 0.0455 & 0.0449 & -0.7604 & 0.0021 & 0.0022 & 0.0132 \end{bmatrix},$$

$$Q_2 = \begin{bmatrix} 91.1962 & -4.9524 & -7.3058 & -0.6946 & -0.0137 & 0.0401 \\ -4.9524 & 64.8688 & -6.0623 & 0.0279 & -0.4145 & 0.0431 \\ -7.3058 & -6.0623 & 97.5787 & 0.0285 & -0.0116 & -0.7758 \\ -0.6946 & 0.0279 & 0.0285 & 0.0115 & 0.0022 & 0.0021 \\ -0.0137 & -0.4145 & -0.0116 & 0.0022 & 0.0049 & 0.0022 \\ 0.0401 & 0.0431 & -0.7758 & 0.0021 & 0.0022 & 0.0136 \end{bmatrix},$$

$$Q_3 = \begin{bmatrix} 59.0632 & -5.0682 & -6.3968 & -0.3917 & 0.0049 & 0.0337 \\ -5.0682 & 60.8165 & -6.5345 & 0.0001 & -0.4149 & 0.0331 \\ -6.3968 & -6.5345 & 69.7799 & -0.0019 & 0.0018 & -0.5558 \\ -0.3917 & 0.0001 & -0.0019 & 0.0067 & 0.0019 & 0.0021 \\ 0.0049 & -0.4149 & 0.0018 & 0.0019 & 0.0073 & 0.0021 \\ 0.0337 & 0.0331 & -0.5558 & 0.0021 & 0.0021 & 0.0111 \end{bmatrix},$$

$$Q_4 = \begin{bmatrix} 49.1919 & -3.3174 & -4.2157 & -0.4438 & 0.0021 & 0.0252 \\ -3.3174 & 42.6652 & -4.0112 & 0.0224 & -0.3625 & 0.0355 \\ -4.2157 & -4.0112 & 52.8587 & 0.0099 & 0.0035 & -0.4967 \\ -0.4438 & 0.0224 & 0.0099 & 0.0091 & 0.0024 & 0.0025 \\ 0.0021 & -0.3625 & 0.0035 & 0.0024 & 0.0064 & 0.0023 \\ 0.0252 & 0.0355 & -0.4967 & 0.0025 & 0.0023 & 0.0108 \end{bmatrix},$$

$$O_1 = \begin{bmatrix} -12574 & 770 & 814 \\ 770 & -12598 & 1059 \\ 814 & 1059 & -12769 \end{bmatrix},$$

$$O_2 = \begin{bmatrix} -12490 & 827 & 415 \\ 827 & -12595 & 935 \\ 415 & 935 & -12570 \end{bmatrix},$$

$$O_3 = \begin{bmatrix} -8890.5 & 163.7 & 316.1 \\ 163.7 & -8865.5 & 320.6 \\ 316.1 & 320.6 & -8664.8 \end{bmatrix},$$

$$O_4 = \begin{bmatrix} -5896.2 & 193.5 & 128.1 \\ 193.5 & -5985.4 & 232.8 \\ 128.1 & 232.8 & -5904.0 \end{bmatrix},$$

$$L_1 = \begin{bmatrix} 370160 & -18210 & -37230 \\ -5120 & 309560 & -24930 \\ -62930 & -60550 & 469170 \\ -2510 & 40 & 50 \\ -110 & -1940 & -80 \\ 430 & 440 & -3670 \end{bmatrix}, L_2 = \begin{bmatrix} 437770 & -41530 & -42600 \\ -14220 & 310500 & -22000 \\ -49140 & -53640 & 469420 \\ -3330 & 280 & 190 \\ -120 & -1960 & -100 \\ 300 & 410 & -3730 \end{bmatrix},$$

$$L_3 = \begin{bmatrix} 186440 & -18190 & -23090 \\ -20060 & 190580 & -25500 \\ -34660 & -35090 & 208300 \\ -1220 & 30 & 30 \\ 50 & -1290 & 50 \\ 220 & 220 & -1660 \end{bmatrix}, L_4 = \begin{bmatrix} 105470 & -11190 & -10920 \\ -9370 & 93370 & -11390 \\ -12080 & -14630 & 113900 \\ -950 & 90 & 40 \\ 30 & -790 & 30 \\ 80 & 130 & -1070 \end{bmatrix}.$$

Compared the adaptive algorithm in [42], the FAFE algorithm used in the adaptive observer has obvious advantages, which can be seen in Figure 6.

From Figure 6, it can be obviously seen that both observers have an accurate estimation of faults and disturbances. The observer method proposed in this paper has a shorter response time and smaller steady-state error than the method proposed in [42].

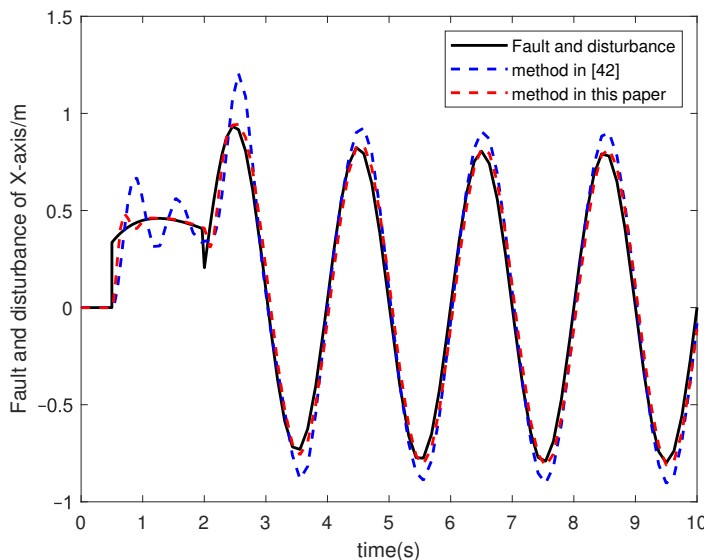

**Figure 6.** Estimated fault and disturbance from observer.

From Figures 7 and 8, the method proposed by this paper can do well with actuator faults and disturbances, and finally converge to zero.

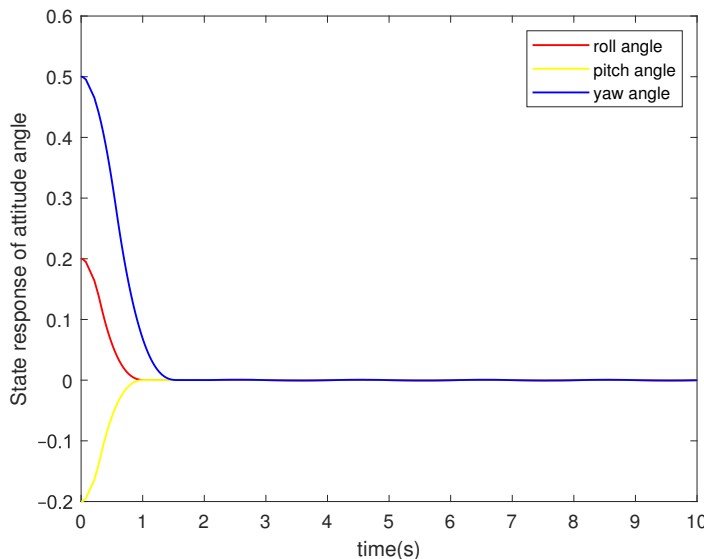

**Figure 7.** State response of attitude angle.

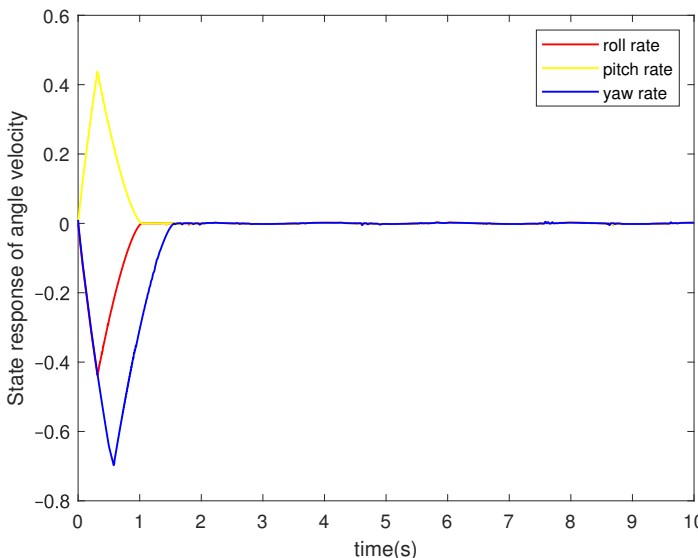

**Figure 8.** State response of angular velocity.

Figures 9 and 10 show the state response of attitude angle and angle velocity, based on [43]. The state responses in Figures 7 and 8 have faster responses than those in Figures 9 and 10. Comparing the results of the two methods, the nonsingular fast terminal sliding mode controller proposed in this paper has better control performance.

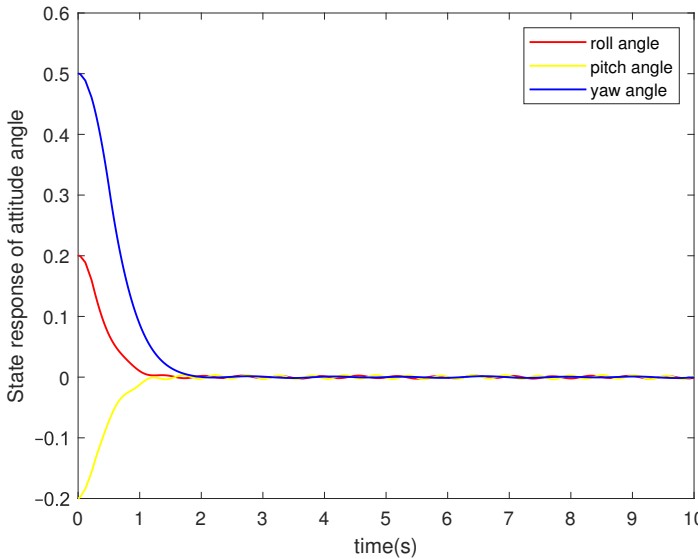

**Figure 9.** State response of the attitude angle by the method in [43].

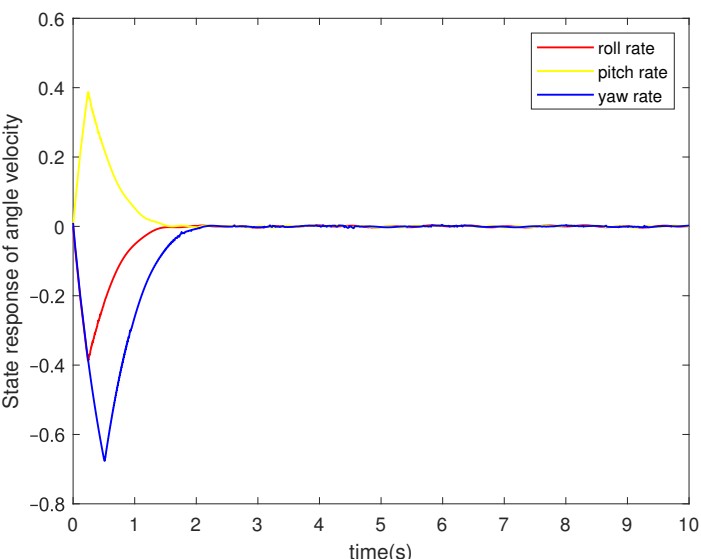

**Figure 10.** State response of the angular velocity by the method in [43].

**5. Conclusions**

In this paper, an adaptive-based nonsingular fast terminal sliding mode controller of Markov jump nonlinear systems is discussed. An adaptive observer with the FAFE method is proposed to estimate the actuator faults and external disturbances. Based on this observer, a nonsingular fast terminal sliding mode controller is provided for fault-tolerant control. By utilizing the Lyapunov-Krasovskii functional (LKF) and linear matrix inequalities (LMI) techniques, the original system is ensured to be stochastic stable.

Through the method proposed in this paper, the MJNSs can demonstrate resistance to actuator faults and disturbances. In practice, UAVs may encounter more complex situations, including human factors and environmental factors. Future research should consider more factors and focus on eliminating chattering.

**Author Contributions:** Conceptualization, P.Y.; methodology, P.Y. and Z.S.; validation, Z.S.; formal analysis, Z.S.; investigation, Y.D. and K.F.; writing—original draft preparation, Z.S.; writing—review and editing, Z.S., Y.D. and K.F.; supervision, P.Y. All authors have read and agreed to the published version of the manuscript.

**Funding:** This work was funded by Key Laboratories for National Defense Science and Technology (6142605200402), National Key Laboratory of Science and Technology on Helicopter Transmission (No.HTL-O-21G11), the Aeronautical Science Foundation of China (20200007018001), the Aero Engine Corporation of China Industry—university-research cooperation project (HFZL2020CXY011), and the Research Fund of State Key Laboratory of Mechanics and Control of Mechanical Structures (Nanjing University of Aeronautics and Astronautics) (MCMS-I-0121G03).

**Institutional Review Board Statement:** Not applicable.

**Informed Consent Statement:** Not applicable.

**Data Availability Statement:** Not applicable.

**Conflicts of Interest:** The authors declare no conflicts of interest.

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
