# Peer review of "Fast Terminal Sliding Mode Fault-Tolerant Control for Markov Jump Nonlinear Systems Based on an Adaptive Observer"

_drones, doi:10.3390/drones6090233_

Round 1
Reviewer 1 Report
The authors discuss the problem of observer design to estimate the actuator fault and disturbance of an UAV. The topic is for interest for the journal readers. The paper is well structured. The reference list is too restrictive. It must be updated with more papers in the fault estimation and UAV control field. The paper needs proofreading (especially typos must be corrected). The main issue of the paper is the lack of practical implementation (or even discussion). For example what practical limitations arise from the two assumptions? On which measurements is based the proposed observer? How the parameters from Table 1 are transposed in the system model? A series of work demonstrated that the use of quaternions is beneficial (e.g. https://doi.org/10.3390/math8101829). Please discuss in your case. To use such observer online what kind of equipment is needed (LMI solving online)?
Reviewer 2 Report
attached

Reviewer 3 Report
Dear editors, dear authors,
my main concern is that this work has very little to do with drones. It is true that the simulations present a (simplified) UAV model, but the point is that it you really want to the make the work suited for "Drones", this manuscript requires a complete rewriting, in order to highlight the drone application from the very beginning.
Yet, focusing on the simulations, the authors do not make it clear what is the meaning of "transition rates matrix" in a drone. I guess that the authors assume that faults occur intermittently from one actuator to the other. I am not sure this is a very realistic scenario.
Also, the authors present the angle and angle velocity, but never present the control input in their simulations, which is also not very realistic
Round 2
Reviewer 1 Report
The authors revised the paper in accordance with the major comments. The practical implementation issues remains unsolved. I understand that the paper is based on simulations, but these simulations must be relevant for further practical implementation. At least the response for the reviewer must be included in the manuscript.
Reviewer 2 Report
The authors stated that they tried best to improve the manuscript and made some changes in the manuscript. To the regret, their changes did not influence the content and framework of the paper that left allmost all remarks to the work, related with proposed stochastic control analysis not taken into account. Thereby, the author's key point based on the fault-tolerate needs many proofs and corrections, the work in the present form does not meet approval.
Reviewer 3 Report
Dear editor dear authors, it seems that the editor is relatively happy with this work and therefore a new review round was allowed. No problem, I would just the authors to be aware of recent works on UAVs with adaptive control and sliding mode (also called adaptive sliding mode): An underactuated control system design for adaptive autopilot of fixed-wing drones; Adaptive vector field guidance without a priori knowledge of course dynamics and wind. The reason is that adaptive sliding mode can deal with uncertainty in the system dynamics including faults and disturbances. This seems to be in line with the goal of this manuscript.
